# Rethinking the Subjective Units of Distress Scale: Validity and Clinical Utility of the SUDS

**DOI:** 10.3390/clinpract15070123

**Published:** 2025-06-29

**Authors:** Elizabeth Mattera, Brian Zaboski

**Affiliations:** Yale School of Medicine, Department of Psychiatry, Yale University, New Haven, CT 06510, USA; elizabeth.mattera@yale.edu

**Keywords:** subjective units of distress scale, SUDS, validity, construct validation, psychometric properties, clinical assessment, affect measurement, exposure therapy

## Abstract

The Subjective Units of Distress Scale (SUDS) is a widely used self-report measure clinicians rely on during exposure and response prevention (ERP) to monitor progress, guide exposure pacing, and assess intervention efficacy. However, despite its ubiquity in clinical and research settings, foundational investigations of its psychometrics are often atheoretical, fail to evaluate its longitudinal properties, and lack a rigorous construct validation framework. This paper addresses these shortcomings by evaluating the SUDS as a measure of state negative affective intensity using the Strong Program of Construct Validation. Our evaluation demonstrates that the SUDS suffers from significant psychometric weaknesses, including construct underrepresentation, construct irrelevance, poorly defined measurement occasions, and structural limitations, challenging its validity as a precise measure of subjective distress. These limitations have crucial implications for clinical practice, potentially leading to misinterpretations of patient distress and compromising treatment decisions. We discuss these clinical implications, highlight them with a brief clinical vignette, outline a research roadmap for potential improvement using modern psychometric methods, and provide practical recommendations for clinicians currently using the SUDS. Given these validity concerns, caution is warranted when interpreting SUDS scores in both clinical and research contexts until its psychometric properties are more robustly established and understood.

## 1. Introduction

Wolpe and Lazarus introduced the Subjective Units of Distress Scale (SUDS) in 1966 [1]. They described the administration of the SUDS through the following interaction with a hypothetical patient.

Think of the worst anxiety you have ever experienced, or can imagine experiencing, and assign to this the number 100. Now think of the state of being absolutely calm and call this zero. Now you have a scale. On this scale how do you rate yourself at this moment? (p. 73).

Wolpe referred to each unit of measurement as a “sud” (p. 73), or a subjective unit of disturbance, with which patients communicated the severity of their distress throughout behavior therapy. In 1981, Wolpe and Wolpe [2] anchored the SUDS with the descriptive terms displayed in Table 1.

Consequently, the SUDS afforded practical advantages in clinical and research settings. First, it collected within- and between-session ratings in a dynamic therapeutic environment across disorders. In recent studies, the SUDS has been used at the start and end of sessions [3], at multiple timepoints throughout sessions, and immediately before and after exposures [4]. Second, it allowed clinicians to rank-order fears on a hierarchy, a graduated list of anxiety-provoking situations that frequently guided exposure-based treatment [5]. Third, it helped prevent exposure-based treatment from becoming too overwhelming. For instance, if a patient reported a SUD for one exposure as a 20 but the next as an 80, a clinician could develop intermediate challenges that progressively intensified the experience [6]. Thus, SUDS scores directly influenced critical clinical decisions, such as determining the pace of exposure therapy, assessing treatment readiness, judging session effectiveness, and even defining treatment success (e.g., the common 50% reduction heuristic for habituation) [7].

With a convenient measure applicable to clinical and research settings, the SUDS was swiftly integrated into anxiety research throughout the 1960s and 1970s. Pioneers in anxiety research found that within- and between-session decreases in the SUDS related to habituation during the treatment of OCD and agoraphobia [8], while dissertations using the SUDS showed higher distress for teachers exposed to unruly classroom behaviors than teachers who were not [9]. Wolpe himself argued for the SUDS as both a clinical tool [6] as well as a research tool, which he used widely in his own seminal investigations in exposure work [10].

The SUDS continues to be used in modern psychology, primarily for measuring state negative affective intensity within and between therapy sessions [11,12] and as a measure of subjective fear, anxiety, and discomfort [13,14] across various settings [15,16]. While other scales are often interminable, costly, and measure trait characteristics (vs. states), the SUDS provides a method of measuring unstable constructs like distress and anxiety in real time [17]. In several training and practitioner-friendly resources, the SUDS continues to be the de facto method of generating modern fear hierarchies [7,18,19]. Indeed, the SUDS, along with a 50% reduction cutoff, is often used to signify that habituation to a feared stimulus has occurred [20,21]. Clinicians rely on these SUDS ratings moment-to-moment to gauge patient tolerance, decide whether to continue, intensify, or cease an exposure task, and to collaboratively build fear hierarchies that form the backbone of treatment plans [20]. The SUDS has also played an important role in basic and applied research clarifying exposure therapy’s mechanistic underpinnings [22]. For example, in a study concluding that habituation is a separate phenomenon from extinction learning, SUDS ratings of conditioned stimuli were one of the measures [23]. Because it is such a popular and ubiquitous measure, a careful examination of its construct validity is warranted.

In clinical practice, accurate measurement of patient distress is fundamental to treatment planning, progress monitoring, and outcome evaluation. The SUDS has been widely adopted in exposure-based treatments across anxiety disorders, PTSD, and OCD, where clinicians rely on it to calibrate exposure intensity, determine habituation, and make moment-to-moment treatment decisions. However, as Messick (1995) [24] and Cronbach and Meehl (1955) [25] established, meaningful clinical application requires construct clarity before empirical utility can be established. While the SUDS has been a subject of various investigations over the decades, a comprehensive assessment through a rigorous, modern construct validation lens has been less common, which is the gap our paper seeks to address. Thus, our narrative and theoretical critique through the rigorous Strong Program of Construct Validation [26] addresses a critical gap affecting everyday clinical practice—namely, that clinicians using the SUDS to guide treatment decisions may be measuring qualitatively different phenomena across patients, sessions, or even within the same exposure exercise. This theoretical ambiguity undermines treatment standardization, complicates clinical training, and potentially compromises patient outcomes. Therefore, the primary objective of this paper is to critically reevaluate the construct validity of the SUDS using the rigorous framework of the Strong Program of Construct Validation. We will argue that this examination reveals significant foundational weaknesses that have profound implications for its use and for future research directions. Our goal is to provide the theoretical groundwork necessary for improving this widely used measure or developing sound alternatives. For convenience, a glossary of terms is provided.

## 2. Validity Studies

Construct validation—the “integrative and evaluative judgement” that “supports the adequacy and appropriateness of inferences and actions based on test scores” is a necessary step for establishing the interpretation of scores [27] (p. 13). As it relates to the SUDS, three important studies examined construct validity directly. Table 2 summarizes the key findings, interpretations, and our critiques of the seminal studies discussed. 

The earliest attempt explored the relationship between the SUDS, digit temperature, and heart rate in twenty college students [28]. Participants watched a 3-minute video of a venous cutdown procedure, during which SUDs scores were obtained at 30-second intervals. Average heart rate and left-hand and right-hand temperature correlated significantly with SUDs scores. The authors concluded that peripheral vasoconstriction is more sensitive to subjective anxiety than heart rate, and that the results support the continued use of the SUDS in clinical and research settings.

Nevertheless, this overlooks the possibility that the SUDS was measuring other constructs: participants may have reported anxiety, disgust, or general autonomic arousal after watching the video of the venous cutdown procedure. The authors reported that the relationship between the left-hand and right-hand correlations and the SUDS was in the predicted direction. But in the absence of theory, their prediction could still have been confirmed if they found a statistically significant inverse correlation between digit temperature and anxiety (especially since at the time of publication the direction of the relationship was not yet established). Their belief in the SUDS and its predictive ability is plainly stated: “Self-report data such as the subjective anxiety scale are frequently the primary outcome measure of interest in clinical behavior therapy and its usefulness is not dependent on concurrent response parameters” (p. 6). 

In another attempt to study the SUDS, Kim et al. [29] analyzed SUDS scores from 61 patients treated with eye movement desensitization and reprocessing (EMDR) at a trauma clinic. At baseline, the authors stated that the SUDS correlated with the Beck Depression Inventory (*r* = 0.28, *p* < 0.05; [31]) and the State Anxiety Scale (*r* = 0.31, *p* < 0.05), which they argued was evidence for convergent validity. They interpreted a non-significant correlation with the Trait Anxiety Scale (*r* = 0.21, *p* > 0.05; [32]) as evidence for discriminant validity. They further claimed there were no correlations between the SUDS and age (*r* = −0.23, *p* < 0.05), education (*r* = −0.016, *p* < 0.05), or income (*r* = 0.12, *p* < 0.05)—interpreted as discriminant validity. In terms of predictive validity, they added that the SUDS at the end of the first session was significantly and positively correlated with the Clinical Global Impression of Change (*r* = 0.32; CGI-C; [33]). The SUDS also correlated with CGI-C at the end of the second (*r* = 0.51) and third sessions (*r* = 0.61). The authors found significant, positive correlations between the SUDS and the Symptom Checklist-90-Revised’s Positive Symptom Distress Index (SCL-90 PSDI [34] and Impact of Event Scale-Revised (IES-R [35]), both interpreted as evidence of concurrent validity (*r*s = 0.50 and 0.46, respectively).

However, many validity interpretations were based on whether a parameter estimate was statistically significant. Unfortunately, this use of *p*-values is problematic, as it relies on dichotomous decisions that overlook the strength and theoretical importance of the correlations themselves [36]. For instance, the authors concluded that the SUDS did not correlate with the Trait Anxiety Scale because the *p*-value was not significant—which they interpreted as evidence of discriminant validity—when in fact the size of the correlation with the Trait Anxiety Scale (*r* = 0.21) was akin to that with the Beck Depression Inventory (*r* = 0.28), which they did interpret as evidence for convergent validity. Because *p*-values offer limited information about the actual magnitude of an effect, their determination of convergent or discriminant validity based primarily on *p*-value thresholds, without adequate consideration of these comparable correlation strengths (e.g., *r* = 0.21 vs. *r* = 0.28), is inconsistent. While laudable, this investigation lacks a strong foundation for clinicians looking to trust SUDS scores as specific indicators of state distress versus related constructs like depression or stable anxiety traits, highlighting the risk of relying on assessment approaches not grounded in robust, theoretically coherent validation evidence, a necessary step to distinguish evidence-based practice from potentially unsubstantiated claims [37].

In the absence of a theoretical and empirical understanding of the SUDS, its measurement limitations, and the construct it purports to measure, one cannot interpret positive correlations between anxiety and depression as convergent validity. Again, using *p*-values, Kim et al. [29] claimed that neither age, education, nor income was significant, so these were evidence of discriminant validity. Considering predictive validity, the authors noted that SUDS scores following the first session predicted SUDS scores during all subsequent sessions. Nevertheless, these results may all have been artifacts resulting from their last observation carried forward (LOCF) method, which typically involves copying subject data forward in a dataset to fill in missing data. This approach is problematic enough that researchers and statisticians have been skeptical of it for some time [38]; hence, several recommendations have been put forth for more sophisticated analyses/designs [39,40].

To expand the scope of the SUDS’s clinical utility, Tanner (2012) [30] investigated emotional and physical discomfort. They used the Minnesota Multiphasic Personality Inventory-2 (MMPI-2 [41]) and Global Assessment of Functioning (GAF [42]) with the SUDS to track 182 patients in a hospital setting. They found that emotional SUDS scores were moderately related to clinician GAF ratings (*r* = −0.44), Scale A of the MMPI-2 (*r* = 0.35), and the sum of scales 1–3 (*r* = 0.37), and decreased significantly after 3 months of treatment. Physical SUDS scores did not decrease significantly after treatment. The researchers concluded that “the data provide several pieces of evidence regarding the validity and sensitivity of global SUDS ratings” (p. 33) and that global SUDS ratings are a useful extension of the traditional SUDS scale. 

Nevertheless, the foundation for Tanner’s (2012) [30] argument was the work done by Thyer et al. [28] and Kim et al. [29], and they supplied several papers that have found correlations between the SUDS and other constructs. What do we make of the correlations between the SUDS and so many constructs, which have continued to be demonstrated in numerous papers since then? First, we acknowledge that the SUDS is associated with measurements of stress and anxiety, including heart rate and heart variability [43]. It correlates with several constructs and instruments, including willingness to engage in treatment (*r =* −0.12; [44]), Screen for Anxiety and Related Disorders—Youth and Parent reports [45], the externalizing score on the parent report for the Strength and Difficulties Questionnaire [46], and the Global Distress Tolerance Scale [47,48]. Moreover, some research shows that the SUDS predicts between-session exposure therapy treatment outcomes [49], and that changes in SUDS scores can predict changes in outcome variables such as OCD severity, functional impairment, and clinician-rated improvement [50].

Validity evidence requires more than just a collection of correlations—it also requires theoretical justification [25,51]. Thus, a deeper examination of its construct validity is overdue. To systematically evaluate these psychometric concerns and provide a clearer path forward, we use the Strong Program of Construct Validation [26,52]. This is a rigorous, theory-driven framework that helps researchers assess whether a measure, like the SUDS, truly captures the concept it intends to measure by examining different types of evidence in a structured way. We will explore this program and then apply its components—Substantive, Structural, and External—to the SUDS. This framework of construct validation has been used extensively for several decades and is recommended by researchers and modern factor analytic texts [52,53]. Its primary purpose is to help researchers determine if a measure belongs within a nomological network, or a related network of constructs and variables [25]. So it is within this framework that one can evaluate whether enough research and theoretical work suggest that the SUDS is a suitable measure of distress. Our assessment is followed by recommendations for research and clinical practice.

## 3. Strong and Weak Construct Validation

The Weak Program of construct validation is an exploratory endeavor that captures the correlational relationship between the focal construct and other constructs with little regard for theory [51]. Research falls into the Weak Program when it attempts to establish external validity (i.e., relationships with other variables) without first adequately addressing the substantive aspects of construct validation, such as clearly defining the theoretical construct, ensuring the measure comprehensively represents it, and minimizing construct-irrelevant variance. Because the Weak Program relies on exploratory (rather than confirmatory) empirical research, often bypassing these crucial initial theoretical steps, its atheoretical and unsystematic approach provides less convincing support for a construct. This can lead to a collection of correlations that, while perhaps statistically significant, offer little clarity on what the measure truly assesses or how it functions within a coherent theoretical system.

To address the limitations of the Weak Program of construct validation, the Strong Program is a theory-driven framework derived from Loevinger (1957) [54] and Nunnally (1978) [55]. It integrates the six categories of construct validation from Messick’s (1995) [24] unified concept of validity and includes three components: Substantive, Structural, and External. A variety of evidence is required at all stages, which build upon each other. If one stage lacks evidence, it must be reevaluated. Figure 1 displays these components as building blocks.

## 4. Substantive Component

Within the Substantive Component, the theoretical and empirical domains of a construct are identified. A construct’s theoretical domain consists of all that is known about the construct in the literature and is supplemented by the researcher’s observations [26]. Ideally, the theoretical domain corresponds to a construct’s empirical domain, or all ways that the construct can be operationalized (e.g., brain scans, electrophysiological recordings, behavioral measures, self-reports, etc.). Two threats to construct validity in this stage include construct underrepresentation and construct irrelevance [24]. Construct underrepresentation occurs when measures do not sufficiently account for and represent the theoretical domain of the focal construct [56]. Construct irrelevance occurs when an assessment captures variability unrelated to the construct’s theoretical domain. 

A clinician observing a high SUDS score cannot be certain what specific aspect of the patient’s experience it reflects (e.g., fear of the stimulus, general overwhelm, frustration), potentially leading to misinterpretations of the patient’s state and misguided clinical interventions. This is highlighted by applying the Strong Program to “distress.” Without explicitly defining the domain (distress vs. anxiety), it is unclear what the SUDS measures. For instance, patients are supposed to report the “worst *anxiety*” [1] (p. 66) they ever experienced. But it is called a subjective unit of disturbance scale and was also encouraged to be used for domains such as “rejection,” “guilt,” “or others like them” (p. 67). This introduced many constructs—anxiety, rejection, guilt, or others—into the theoretical domain, ostensibly under the umbrella term “disturbance.” Yet, without an operational definition of disturbance and how that relates to anxiety, the scale risks measuring multiple domains simultaneously and introducing construct-irrelevant variance. Even if "distress" is considered broadly as a form of state negative affect, the SUDS fails to adequately define its precise scope within this larger dimension. It does not clarify which specific facets of negative affect are being targeted (e.g., fear, sadness, anger, general unease), nor how it differentiates a global sense of negative feeling from more specific emotional experiences. This lack of specificity makes it difficult to ascertain if the SUDS is truly capturing a coherent aspect of negative affect or simply an amalgamation of various undifferentiated negative feelings, further contributing to construct irrelevance. For instance, someone with both anxiety and depression may have a high SUDS score because of depression, anxiety, or other reasons. By integrating multiple constructs of interest, we risk combining the intensity of different affective responses.

The SUDS also fails to define its measurement period, obscuring the construct it purports to measure. Inconsistent prompting (rate your distress now vs. rate your peak distress during that exposure) across or within sessions can yield incomparable data, undermining the clinician’s ability to accurately track change or therapeutic processes like habituation. For instance, in an investigation of public speaking anxiety, the SUDS was a measure of distress throughout the course of a speech [57]. In another study measuring eating disorder treatment, the SUDS evaluated distress prior to and after eating a feared food, during within-session and between-session habituation [58]. Different domains measured on different time scales are subject to their own theoretical and empirical examinations. Each of these, as a measure of state negative affective intensity, requires a theoretical and empirical understanding.

## 5. Structural Component

When enough evidence is collected in the Substantive Stage, a construct should be evaluated in the Structural Stage. Here the relationship between the observed variables and the construct of interest is assessed [26]. Constructs assessed by single-item measures are best assessed by their sensitivity to within-person, longitudinal variability [59,60]. In other words, fluctuations in one person’s scores may be a better measure of their affect than comparing their scores to others’ scores. Further, measures of affect must be considered in the context of an individual’s traits [61]. For instance, while significant fluctuations in affect scores may be unusual for one person, they may be expected in an individual with greater emotional lability. More concretely, someone with OCD may report consistently high anxiety (low within-person variability), but someone with borderline personality disorder may experience a wider range of emotional fluctuation (greater within-person variability).

The SUDS falls short without clear theoretical and empirical operational definitions in the Structural Component. For instance, the within-person variability of distress as measured by the SUDS may differ depending on whether someone experiences anxiety, “Not just right,” or hopelessness. Relying on a single SUDS number may cause clinicians to miss crucial patterns or sources of distress variability (e.g., lability vs. stable high anxiety), hindering a nuanced understanding of the patient’s affective state and response to treatment. Also, the construct’s irrelevant variance in the Substantive Component carries through to the Structural Component. For example, a score after an exposure therapy session may be contaminated by the many different constructs the SUDS is measuring. Consequently, the extent and the source of within-person variability remain unexplained.

## 6. External Component

In the External Component, an individual’s time series of momentary affective scores would be compared to trait measures [61]. For instance, we may hypothesize that average longitudinal scores of anxiety and the degree of variation in reported anxiety over time would correlate with Beck Anxiety Inventory scores (BAI [62]). We might also hypothesize that a focal construct should change in the context of group differences [63]. For instance, a researcher might randomize children to either a treatment or experimental condition and provide cognitive–behavioral therapy, hypothesizing that therapy will reduce scores on the SUDS in the treatment group. Similarly, a researcher could use knowledge of the construct to predict how different populations of individuals (OCD vs. non-OCD) might rate the SUDS in the presence of certain stimuli.

However, because the SUDS’ focal construct is not theoretically or empirically defined in the Substantive Stage—leaving its evidence from the Structural Stage undetermined—evidence from the External Stage is difficult to interpret. Without this theoretical and empirical knowledge, one cannot know, for example, how to make sense of the relationships described by Kim et al. [29], who found discriminant validity on the grounds of a non-significant correlation between the SUDS and Trait Anxiety Scale, and then found convergent validity when the SUDS correlated with the BDI. More broadly, without a clear understanding of what the SUDS measures, clinicians cannot confidently use it to predict treatment response, differentiate patient groups, or meaningfully relate subjective distress to other clinical outcomes. For this reason, Figure 1 suggests that conceptual work be completed in earlier stages in the Strong Program.

## 7. Discussion

Since its development, the SUDS has been a cornerstone of research and clinical practice, providing a quick and convenient method of measuring subjective anxiety and distress [64]. To evaluate the utility of the SUDS as a measure of distress, we investigated its development and applied the Strong Program of Construct Validation, a framework that outlines three cumulative and recursive steps for establishing evidence of construct validity: Substantive, Structural, and External. 

A brief clinical vignette can illustrate how a failure at the Substantive Stage—specifically, a lack of clear construct definition for “distress”—can manifest in practice: A patient with OCD undergoing ERP for contamination fears consistently reports a SUDS score of 80/100 during exposures involving touching a contaminated object. The clinician, following standard practice, waits for the SUDS score to decrease by 50% before ending the exposure, but the patient’s rating remains high. Based solely on the SUDS, the clinician might conclude the exposure is ineffective or too difficult. However, upon further questioning prompted by the lack of observable avoidance reduction, the patient reveals the high rating reflects intense frustration (“I should be over this!”), hopelessness (“This is never going to work”), and even their anxiety about their progress and engagement in ERP (“Am I doing this right? What if I’m untreatable?”), rather than acute anxiety about contamination itself. The SUDS score, lacking clear construct definition (Substantive failure), provided misleading information. Relying on it obscured other clinically relevant affective states and potentially stalled effective treatment progression, underscoring the critical importance of precise differential assessment when interpreting subjective reports in clinical practice [65].

With an understanding of where the SUDS stands in terms of the Strong Program, its place in modern assessment, as well as how to improve it, becomes clearer. The SUDS struggled to recover from the limitations in its development. For example, Wolpe and Lazarus [1] approached depression and anxiety from a combined theoretical framework in their formulation of the SUDS. They asserted that “most neurotic depression is the product of severe anxiety arousal” (p. 28), potentially illuminating why the SUDS measures so many constructs. At the same time, little was known about how to measure constructs adequately over longitudinal periods [66]. Nevertheless, the process of validation is ongoing [56], and the SUDS should still meet modern measurement standards for clinical and research decisions.

## 8. Clinical Implications

Although the practical appeal of the SUDS in demanding clinical settings—offering a seemingly rapid, quantifiable snapshot of patient distress—is undeniable, the significant psychometric limitations regarding its construct validity necessitate considerable caution from clinicians relying on it. Instead of accepting the SUDS score at face value, it is advisable to supplement it with other data sources. Clinicians should integrate direct behavioral observations, such as latency to engage in exposure, task duration, avoidance attempts, or non-verbal signs of distress, alongside the SUDS rating. Crucially, employing qualitative inquiry can contextualize the number; asking clarifying questions like “What specific feelings or thoughts are contributing to that rating right now?” may reveal whether the score reflects the targeted anxiety, or perhaps frustration, hopelessness, or physical discomfort, thus preventing potential misinterpretations. 

Furthermore, clinicians could attempt to improve specificity by explicitly defining the construct during administration (e.g., “Rate your fear of contamination from 0–100 right now”) rather than using ambiguous terms like “distress” or “disturbance.” Furthermore, adopting more structured anchoring procedures, as exemplified in some treatment protocols like Prolonged Exposure for PTSD, where therapists help clients generate specific, personalized examples for various SUDS levels (e.g., 0, 25, 50, 75, 100) that remain consistent throughout therapy [67], might also enhance clarity and consistency. While validated single-item measures for specific momentary affective states are being developed [52,53], psychometric robustness for guiding real-time clinical decisions within therapy sessions requires further dedicated research. For evaluating change across sessions or treatment phases, clinicians should continue to prioritize validated multi-item questionnaires designed to assess specific, relevant constructs beyond momentary distress, such as quality of life or symptom severity [68]. The overarching goal must be to base clinical judgments and interventions on measures with demonstrated reliability and validity for the specific inferences being made, a standard the SUDS, in its current form, struggles to meet.

Given the highlighted psychometric limitations of the SUDS, particularly concerning its construct clarity and potential for misinterpretation, clinicians may seek alternatives for assessing and guiding treatment. Better-validated, often multi-item, questionnaires can be used for between-session, within-session, and moment-to-moment improvement. For instance, administration of the PTSD Checklist for DSM-5 [69], prior to exposure therapy for trauma and after treatment, could offer a more interpretive measure of change between sessions. Likewise, the Beck Anxiety Inventory [62] can be used to measure improvements in anxiety, and the Yale–Brown Obsessive–Compulsive Scale [46] can be used to measure change in OCD severity during treatment. These can be used with shorter scales to investigate within-session fluctuations in anxiety, fearfulness, or unease, like the Patient-Reported Outcomes Measurement Information System [70], which now includes computer-adaptive administration. Lastly, these can be combined with moment-to-moment measures that have stronger theoretical support based on the clinical need. For instance, because expectation violation is crucial during modern exposure-based practice [71,72], a 100-point expectancy rating scale (from 0, “will not occur,” to 100, “definitely will occur”) can be used with clients to measure the discrepancy in a client’s prediction between exposures [73]. Utilizing such measures aligns with the foundational principle of construct validation: ensuring that clinical decisions are based on assessments that reliably and validly measure the intended psychological attribute of interest.

## 9. Improving the SUDS

While this evaluation highlights significant concerns regarding the SUDS’ psychometric soundness, it also underscores the genuine need clinicians have for brief, real-time measures of subjective distress during therapeutic interventions like exposure therapy. To bridge this gap and provide clinicians with more trustworthy tools, a dedicated research roadmap is necessary. Firstly, foundational work must revisit the Substantive Component by rigorously defining the specific construct(s) intended for measurement. Rather than relying on the ambiguous umbrella term “distress” or “disturbance,” research employing qualitative methods, such as interviewing patients and clinicians about their moment-to-moment experiences during therapy, could identify and operationalize the most salient affective states (e.g., fear intensity, anxiety, disgust, hopelessness) relevant to specific therapeutic contexts. This clarification is a prerequisite for developing or refining any measurement tool [74]. 

After building a stronger theoretical base, the measurement properties of the SUDS scale warrant empirical scrutiny. How do patients interpret the 0–100 range and the provided anchors (Table 1)? Research could investigate whether the generalized anchors (Table 1) are as effective as more personalized, multi-point anchoring systems used in specific protocols [67]. Exploring the impact of such detailed anchoring on the scale’s consistency and perceived meaning for patients is a worthwhile avenue. Cognitive interviewing studies exploring patients’ thought processes as they generate a SUDS rating, or potentially advanced psychometric analyses like item response theory, could shed light on whether the scale yields meaningful quantitative information [75,76,77]. Furthermore, future validation efforts must move beyond simple atheoretical correlations. Employing intensive longitudinal designs, such as ecological momentary assessment (EMA) within or across sessions, would allow researchers to examine the SUDS’ sensitivity to change and its relationship with specific therapeutic events and validated measures of distinct emotional states over time. Utilizing a multitrait–multimethod approach within these studies would be crucial for systematically establishing the convergent and discriminant validity that has thus far been lacking

EMA research has already been underway for negative affect that falls under the umbrella of distress using the Strong Program of Construct Validation [52]. For example, with specific reports of affective experience, Cloos and colleagues [52] considered momentary affect as individual affective experiences on a dimensional spectrum, ranging from positive affect to negative affect. This underscores that distress is broad, and that measures of negative affect cannot be created on the spot. Thus, we recommend that clinicians and researchers instead operationalize the construct in which they are interested—fear? hopelessness? expectation violation?—and select a validated measure for research and clinical care. Another recommendation is to quantify construct validity through meta-analysis [78]. In attempting to validate a construct, one attempts to include a measure within a nomological network in which it relates to other variables [25]. A meta-analysis can put one’s hypotheses about the SUDS to the test.

## 10. Conclusions

This manuscript has critical implications for researchers and clinicians about a distress scale that has been used for decades. Distress is a uniquely subjective negative affective experience, constantly fluctuating, and influenced by internal and external variables [79]. As the subjective nature of affect remains pertinent [80], it requires varied and modern measurement techniques (e.g., questionnaires, interviews) with an overall sensitivity to discriminant and incremental validity [81]. Modern psychometric methods can supplement such approaches, providing new ways for researchers and clinicians to measure distress that may be applied to the SUDS. By incorporating longitudinal measures of negative affect or using meta-analyses with careful consideration of theory, we can more accurately measure and interpret a rating of distress. 

Given the significant unanswered questions about what the SUDS measures and the potential for misinterpretation, we urge clinicians and researchers to be highly cautious and critical when employing it. While its simplicity is appealing, relying on SUDS scores without acknowledging their profound psychometric limitations may compromise clinical decision-making. However, developing and validating brief, reliable, and theoretically grounded measures of subjective state distress suitable for the dynamic context of therapy sessions remains an important and attainable goal to better support evidence-based clinical practice.

## Figures and Tables

**Figure 1 clinpract-15-00123-f001:**
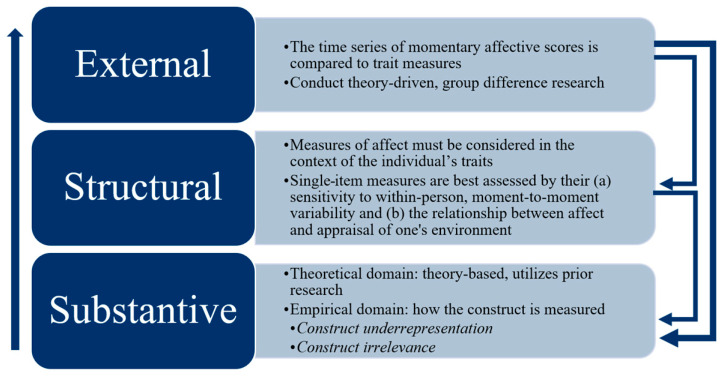
The Strong Program of Construct Validation. Note: Building blocks for the Strong Program. Each block requires a range of theoretical and empirical support. Arrows on the right show that, as support fails in one component, reevaluation is needed in a previous stage. Text boxes highlight useful concepts in each component.

**Table 1 clinpract-15-00123-t001:** Anchors for the subjective units of disturbance scale.

Value	Description
0	No anxiety at all; complete calmness
1–10	Very slight anxiety
10–20	Slight anxiety
20–40	Moderate anxiety; definitely unpleasant feeling
40–60	Severe anxiety; considerable distress
60–80	Severe anxiety; becoming intolerable
80–100	Very severe anxiety; approaching panic

**Table 2 clinpract-15-00123-t002:** Summary of key findings and critiques from validation studies.

Feature	Thyer et al. (1984) [28]	Kim et al. (2009) [29]	Tanner (2012) [30]
Study Focus/Sample	Relationship between SUDS, digit temperature, and heart rate in 20 college students watching a venous cutdown video.	SUDS scores from 61 patients undergoing EMDR at a trauma clinic.	Emotional and physical SUDS in 182 hospital patients, correlated with MMPI-2 and GAF.
Key Correlations	- SUDS and Digit Temperature: Significant, predicted direction. - SUDS and Heart Rate: Significant.	- SUDS and BDI: *r* = 0.28 (*p* < 0.05). - SUDS and State Anxiety: *r* = 0.31 (*p* < 0.05). - SUDS and Trait Anxiety: *r* = 0.21 (*p* > 0.05). - SUDS and Age: *r* = −0.23 (*p* < 0.05). - SUDS and Income: *r* = 0.12 (*p* < 0.05). - SUDS and SCL-90 PSDI: *r*s = 0.50. - SUDS and IES-R: *r*s = 0.46.	- Emotional SUDS and GAF: *r* = −0.44. - Emotional SUDS and MMPI-2 Scale A: *r* = 0.35. - Emotional SUDS and MMPI-2 Scales 1–3: *r* = 0.37. - Emotional SUDS decreased significantly over 3 months.
Authors’ Interpretation	Supported continued use of SUDS in clinical and research settings.	- BDI, State Anxiety: Convergent validity. - Trait Anxiety: Discriminant validity. - Age, Education, Income: Claimed “no correlations,” interpreted as discriminant validity. - SCL-90, IES-R: Concurrent validity. - CGI-C Correlations: Predictive validity.	Data provided evidence for validity and sensitivity of global SUDS ratings; a useful extension of traditional SUDS.
Key Limitations	- Overlooked that SUDS might measure constructs other than anxiety (e.g., distress, disgust, general arousal). - Lack of theoretical grounding for predictions. - Asserted SUDS usefulness independently of concurrent physiological measures.	- Validity interpretations relied on statistical significance (*p*-values) rather than effect sizes. - Inconsistently claimed “no correlations” for age, education, and income. - Findings may be artifacts of last observation carried forward (LOCF). - Lacked a strong theoretical basis for interpreting correlations as convergent (e.g., SUDS with depression).	- Argument for validity relies on atheoretical work of Thyer et al. and Kim et al. - Provides no theoretical rationale for SUDS use.

## Data Availability

No new data were created or analyzed in this study.

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
