# Peer review of "Rethinking the Subjective Units of Distress Scale: Validity and Clinical Utility of the SUDS"

_clinpract, 2025, doi:10.3390/clinpract15070123_

Round 1

Reviewer 1 Report

Comments and Suggestions for Authors

This manuscript provides a thorough and critical examination of the Subjective Units of Distress Scale (SUDS), a widely used self-report tool in clinical psychology. The authors apply the Strong Program of Construct Validation to highlight key conceptual and psychometric limitations of the SUDS, offering meaningful implications for both clinical practice and future research. The paper is well-structured, well-referenced, and makes a compelling case for re-evaluating the use of the SUDS in contemporary therapeutic settings.

Strengths

- Theoretical Contribution: The application of the Strong Program of Construct Validation is highly appropriate and adds rigor to the assessment of the SUDS's psychometric foundations.

- Clinical Relevance: The paper clearly articulates the risks of misinterpretation and misapplication of SUDS scores in clinical contexts, which is critical for practicing clinicians.

- Actionable Recommendations: The authors provide a constructive roadmap for future research, including the use of ecological momentary assessment and construct-specific alternatives.

- Clarity and Structure: The manuscript is logically organized and communicates its points effectively, with a balance of theoretical analysis and clinical examples.

Weaknesses and Suggestions for Improvement

  1. Accessibility of Theoretical Framework: While the theoretical analysis is robust, the early sections may be dense for practitioners. Consider briefly summarizing the Strong Program at the outset and expanding on it progressively.
  2. Discussion of Alternatives: The manuscript critiques the SUDS effectively but could benefit from a clearer discussion of validated alternative measures that clinicians could use in the interim.
  3. Empirical Support: While the paper critiques existing validation studies, a table summarizing key findings (e.g., correlations, effect sizes) across studies would enhance clarity and support the central claims.
  4. Clarification of Terminology: The use of “distress,” “anxiety,” and “disturbance” is central to the argument. A glossary or brief definition table could help delineate these terms and improve interpretability.

Reviewer 2 Report

Comments and Suggestions for Authors

The subject of the ms. is interesting and the ms. is clearly written. However, what is missing at both the theoretical and the methods level is the association of the general term of distress with negative affect as a broader dimension of affect than the specific qualities of emotions. PANAS also measures such a broad dimension of affectivity (in fact it measures both negative affect and positive affect) and the information that such measures provide is extremely valuable. In the same vein, SUDS is a valuable instrument in the general form that is administered. On the other hand, the examination of its validity under the Strong Program of Construct Validation seems interesting and methodologically sound. However, here the only thing that is presented is the relative general argumentation and not any trial to examine the instrument empirically.    

Reviewer 3 Report

Comments and Suggestions for Authors

This manuscript is well-structured and it recommends to all who use the Subjective Units of Distress Scale (SUDS) to require only one feeling to be measured by this scale to avoid misinterpretation, and if possible, to use also another measure of the same feeling, to establish scientifically validity of SUDS. In this way, the collected research findings could be used in a meta-analysis that could further clarify the construct that is measured. 

The authors may also check the following source that also uses SUDS in a different way described in the manuscript under review, with anchoring events for a given percentage of the state measured by the scale:

Bluett, E. J., Zoellner, L. A., & Feeny, N. C. (2014). Does change in distress matter? Mechanisms of change in prolonged exposure for PTSD. Journal of behavior therapy and experimental psychiatry45(1), 97–104. https://doi.org/10.1016/j.jbtep.2013.09.003 

Reviewer 4 Report

Comments and Suggestions for Authors

This is an interesting manuscript. Furthermore, it raises some serious questions about the statistical and psychometric qualities and adequacy of a rather popular scale, the SUDS. You present several bibliographical references to support your points and show the inadequacy of the SUDS.

However, two things are not very clear (to me at least):

1) If the SUDS has so limited psychometric properties, why is it still being used?

2) Why go into all these analyses if the SUDS is so limited? Why not just simply say "drop it, it is a poor/outdated tool, use another one".

Finally, how you make the connection between the current state of the SUDS psychometric properties and the introduction of the Weak and (following it) the Strong Construct Validation Program. Perhaps, a better presentation of the weak (although insufficient for the purposes of the SUDS validation) might be useful, so that the introduction to the Strong Construct Validation program outlines more efficiently the limitations of the SUDS.

Reviewer 5 Report

Comments and Suggestions for Authors

General Assessment

This manuscript offers a well-articulated theoretical critique of the Subjective Units of Distress Scale (SUDS), applying the “Strong Program” of construct validation. The work is conceptually rich and raises important clinical concerns. However, its scientific rigor is undermined by several critical weaknesses in its empirical and statistical handling, lack of original data, and at times overly abstract or circular argumentation. 

Abstract and Introduction

The abstract clearly summarizes the manuscript’s purpose: a critical reappraisal of the SUDS scale’s validity through modern construct validation theory. The introduction appropriately outlines the historical role of the SUDS and its current clinical ubiquity.

Points to Improve:

  • The manuscript sets up a straw man by contrasting clinical use with academic rigor without acknowledging recent efforts at psychometric improvement.
  • References are appropriate, but some seminal validation papers are cited uncritically and used as rhetorical devices, not as part of a structured review.
  • No operational hypothesis is stated: although this is a narrative/theoretical review, greater clarity on the objective would enhance readability and testability of claims.

Review of Validity Studies

This is the manuscript’s empirical core, yet it is the weakest part.

Major Issues:

Misuse and misinterpretation of statistical results from prior studies is pervasive. For instance, the critique of Kim et al. (2008) centers on the inconsistent use of p-values but then paradoxically continues to use p-values as a benchmark throughout the paper.

The paper claims that SUDS validity claims are based on dichotomous thinking (p < .05), which is partially true—but this criticism is not supplemented by effect size interpretation, confidence intervals, or meta-analytic synthesis. The authors do not offer any quantitative reanalysis or independent dataset to test these critiques.

LOCF criticism is appropriate, but not contextualized: no alternative method (e.g., multiple imputation, mixed models) is discussed or modeled, making the critique theoretical only.

Recommendation:

Either conduct a systematic review/meta-analysis of SUDS studies or explicitly define this as a narrative/theoretical critique to avoid misleading the reader.

Construct Validation Framework: Strong vs. Weak Programs

This section is well written and pedagogically useful. The explanation of the Strong Program and its application to SUDS is conceptually robust.

Limitations:

Despite being comprehensive, the analysis lacks actual structural validation data (e.g., no factor analysis, no psychometric modeling).

While arguing against the atheoretical use of correlations, the paper does not propose or test a model of SUDS variance across time, settings, or populations. It critiques the field for being too empirical yet contributes no empirical evidence.

Incorporate an exploratory factor analysis or within-subject variability model (e.g., random coefficient models) using existing datasets, or simulate data to demonstrate the points about SUDS inconsistency.

Clinical Vignette and Implications

The clinical example is powerful but anecdotal. It illustrates the key thesis: SUDS scores may not reflect “distress” per se, but frustration, hopelessness, or fatigue.

One example cannot substitute for empirical support. It supports face validity of the authors’ concern, not construct invalidity of the SUDS.

Provide clinician interviews or systematic data (e.g., content analysis of patient ratings) if using qualitative illustrations.

Recommendations and Path Forward

The article makes several sound recommendations:

  • Use of qualitative methods to refine definitions.
  • Adoption of EMA and longitudinal modeling.
  • Development of single-item validated state measures.
  • Meta-analytic construct validation within nomological networks.
  • These are valuable suggestions, but none are undertaken in the current manuscript.
  • The suggestion to abandon SUDS in clinical practice is premature and potentially disruptive without offering a feasible replacement.

Statistical and Methodological Evaluation

Despite the claim to be a rigorous psychometric critique:

  • No new data are presented.
  • No statistical modeling is conducted.
  • No re-analysis or synthesis of existing data is performed.
  • The manuscript fails to meet the standards of either empirical psychometric research or systematic review.

Conclusion: The paper is statistically underpowered, relying heavily on narrative and secondary analysis without original contribution. This is particularly problematic given its critical tone.

As it stands, the article does not meet the methodological standards required for a review or original research paper.

It might be suitable for publication as a position paper or conceptual commentary with appropriate repositioning, less emphasis on statistical rigor, and clear acknowledgment of its limitations.

If the authors wish to retain it as a review or psychometric analysis, they must include:

    • A meta-analytic or systematic review approach.
    • Quantitative assessment of validity across studies.
    • Original data or re-analysis of prior data using modern methods.

Round 2

Reviewer 5 Report

Comments and Suggestions for Authors

Congrats to the authors.